# Risk Assessment Model for the Renewal of Water Distribution Networks: A Practical Approach

**Rodrigo Nunes, Eduardo Arraut and Marcio Pimentel \***

Civil Engineering Division, Instituto Tecnológico de Aeronáutica, São José dos Campos 12228-900, Brazil
* Correspondence: pimentel@ita.br

**Abstract:** Water distribution networks are the most important and costly infrastructure assets of the water supply system, responsible for ensuring a steady and reliable water supply to the end user. Consequently, they are fundamental to the socioeconomic prosperity and health of the population. Therefore, determining pipeline renewal strategies is essential in system management. In this article, the development and application of a simplified risk assessment model allowed to highlight the pipes most susceptible to failures and their respective qualitative (water quality index) and financial consequences in a real case study. The results classified approximately 30 km of the distribution network, highlighting 11 pipes with a high risk of failure (≈3.7 km) and an estimated replacement value of BRL 3.2 million, as a priority for renewal in the next 2 years. In small- and medium-sized water distribution systems with limited technical and financial resources, this model can prove highly useful, as it uses free computer tools and a simple methodology that does not depend on statistical models, mathematical estimates, complex regressions, and intensive computational resources.

**Keywords:** pipe failure consequence; pipe failure probability; pipe renewal strategies; risk component-based model; water distribution networks; pipes and pipelines; rehabilitation; risk and probability analysis; risk assessment; water quality index

## 1. Introduction

Water distribution networks are the most important and costly infrastructure assets of the water supply system, responsible for ensuring that water reaches the end user in adequate quantity and quality. Consequently, these networks are fundamental to the socioeconomic prosperity and health of the population [1–3]. So, the deterioration of this infrastructure presents a great challenge for water utilities around the world. As pipes deteriorate, failures may reduce the hydraulic performance and lead to substantial repair costs, as well as potential dangers of water contamination in the distribution network [4,5]. Therefore, determining pipeline renewal strategies is essential to manage the system [6,7].

A proper distribution network renovation plan should lead to a periodic investigation of the pipelines to determine which ones need to be repaired, renovated, or replaced. Several studies have been carried out so far to determine the most appropriate renewal plans, which classify the rehabilitation activities of the network using deterministic, physical, statistical, and lately, decision-making and risk-based models [8–10]. The deterministic models predict the number of pipe failures or the expected time of the next failure based on the different pipe characteristics and local conditions, and can vary in complexity and form, typically ranging from simple linear regression equations to complex multiple regression equations [11–16]. Ostfeld [17] provided a comprehensive overview of traditional and recent models, including genetic algorithms and ant-colony optimization, as stand-alone or hybrid data driven heuristic or linear/nonlinear optimization schemes. However, many types of data are required for implementation.

Traditional physical models attempt to represent structural processes leading to pipe failure often providing simplified descriptions of physical degradation processes as water

leaks [18,19]. Recently, advanced diagnostic methods have emerged to address these challenges more effectively. One such method, the transient features-based approach [20], leverages hydraulic transient characteristics to detect and predict multi-incipient crack locations, substantially reducing maintenance costs. Another study [21] investigated the impact of gas bubbles on leakage acoustic wave propagation in gas-liquid two-phase flows, highlighting the greater influence of elastic resistance compared to drag resistance. Furthermore, a study on the transient simulation and diagnosis of partial blockages in long-distance water supply systems [22] introduced a diagnostic model based on extreme pressure distribution, demonstrating its effectiveness and suitability for identifying partial blockage zones and intensities in extensive conduits.

Statistical models were developed to address the difficulties of applying physical models and are employed to quantify the structural deterioration of water distribution pipelines based on historical data analyses. However, as with any statistical modeling process, issues related to the "quality and quantity" of this information are crucial for obtaining accurate results [23,24]. Some authors [25,26] compared the strengths, weaknesses, and limitations of these models, as well as problems related to historical data, with general recommendations for overcoming or at least minimizing their occurrence [27].

More recently, studies for the prioritization of pipe renewal have evolved into decision-making models [28–32] and risk assessment models [33–37]. Table 1 shows some of these studies, categorized by type of failure and respective number of risk criteria considered by the authors. Usually, risk is defined as the joint probability of an event occurrence and an estimate of the frequency and physical consequences of undesirable events [38]. This concept applied in water distribution networks refers to the multiplication of the probability of pipe failure and the associated failure consequence, but it is important that these measures are also represented individually as a risk metric in a pipe renewal plan [37,39].

**Table 1.** Some recent studies (2013–2022), categorization (Cat.), and the number of criteria effective in pipe renewal planning.

| Year | Authors | Risk [1] | Pipes Renewal Planning [2] | |
| --- | --- | --- | --- | --- |
| | | | Cat. | No. Criteria |
| 2013 | Devera [34] | PFC | M, H, EC, O, E, S | 10 |
| 2014 | Harvey et al. [40] | PFP | M, O | 5 |
| | St. Clair and Sinha [41] | PFC | M, H, EC, O, QS | 14 |
| 2015 | Kutyłowska [42] | PFP | M | 4 |
| 2017 | Marzouk and Osama [43] | PFC | M, H, EC, O, P | 11 |
| | Winkler et al. [44] | PFP | M, H, O, SL | 10 |
| 2018 | D'Ercole et al. [36] | PFC | H, O | 3 |
| | Salehi et al. [37] | PFP | M, H | 6 |
| 2019 | Phan et al. [45] | PFC | M, EC, O, E | 6 |
| 2021 | Salehi et al. [8] [3] | PFC | M, H, EC, P, S, QS | 19 |
| | Boryczko et al. [46] | PFC | O, S | 2 |
| 2022 | Raspati et al. [47] | PFC | M, H, EC, O | 8 |

Notes: [1]—Risk: PFP = probability of failure pipe; PFC = probability and consequence of failure pipe. [2]—Cat.(main criteria): M = mechanical (material, age, length, diameter); H = hydraulic (pressure, velocity, flow); O = operational (failure history, breakage, leakage); EC = environmental and climate (traffic load, land use, road type); E = economic (repair cost, social cost, lost water); P = physical (soil properties); S = social (customer classification, population without water); SL = spatial (number of valves on the pipe); and QS = qualitative and security (potential of hydraulic/biological threat in the pipe). [3]—Only criteria applied in the case studies used for model validation (Anytown and Two-loop).

Artificial intelligence techniques have also been used in some models for water distribution network renewal, for example: (a) neuro-fuzzy systems for performance evaluation [41] and life cycle cost analysis [43]; (b) artificial neural networks for predicting the time to failure in years [40] and forecasting the failure rate by pipe type [42]; (c) decision tree learning methods to model pipe deterioration [44]. These are purely data-based approaches

that allow for the resolution of complex problems without the need for assumptions of explicitly known and detailed models. However, a large amount of data, programming skills, and computational resources are required, while the internal functioning of the model is not accessible or is only partially known [23].

Given the difficulties of artificial intelligence-based models and considering the type of data typically available, risk assessment models that relate this data to pipe deterioration and associated failure impacts have been considered as an option. For example, Devera [34] developed a simple model that uses the pipe's age and material to calculate the failure probability (P) and the consequences (C) are determined by the proximity of the pipe to heavily traveled roads, critical services, pipe replacement costs, and the pipe flow capacity. The product of these scores, P and C, was represented in a risk matrix that allowed pipes to receive a linguistic classification represented in a geographic information system (GIS) and, consequently, prioritized actions in the system. In addition, D'Ercole et al. [36] used a computer tool that integrates hydraulic simulation with a topological analysis to deal with complex analysis problems. In this approach, the risk assessment is performed by combining the probability of each pipe breaking with the induced consequence in terms of required water demand and pressure deficit. Despite being an alternative to more complex models, these two models depend on historical records of the network, e. g., age/material and break rate.

Salehi et al. [37] presented a hybrid decision-making model based on risk, quantitative, and qualitative analyses, which is independent of operational failure data (often uncertain and/or absent). The method uses design parameters (generally obtained from hydraulic modeling of the system) to evaluate the hydraulic and mechanical conditions of pipes in order to prioritize the rehabilitation actions in water distribution systems. However, it is essential to update these parameters regularly using field data measured. In comparison, Phan et al. [45] developed another hybrid risk-based model to prioritize pipe renewal, in which they proposed a fuzzy inference system to aggregate different topological consequences, using algebraic connectivity (AC) to evaluate the effect of pipe breakage on system performance, considering hydraulic importance, repair cost, and local effects as decision criteria. Although it has been reported that AC has correlations with the hydraulic condition of the distribution system, it was suggested to incorporate the effects of pipe breakages on the hydraulic condition of the network in future studies that develop improvement parameters for economic consequences and effects on public health.

Salehi et al. [8] developed a risk-based component model to determine appropriate pipe renewal strategies in water distribution systems. The developed model is based on a multicriteria decision-making method based on fuzzy logic, where the probability of pipe failure (P) and its respective consequences (C) are analyzed independently. Although the developed model can simultaneously evaluate up to 48 criteria to analyze the risk components of each pipe (P and C), difficulties in implementation have been reported in artificial intelligence-based models. Furthermore, in the model validation, two case studies (Anytown and Two-loop) were considered, where the availability of information limited the number of applied criteria to 19 (categories: hydraulic, mechanical, physical, environmental, social, and safety) and six (categories: hydraulic, mechanical, and safety), respectively.

Boryczko et al. [46] and Raspati et al. [47] focused on risk management in water supply systems. The former emphasizes the importance of preventive strategies and presents a methodology for creating water supply risk maps, using simulations with the EPANET 2.0 software. In contrast, the latter proposes a method that employs the random forest (a machine learning algorithm) and the asset vulnerability analysis toolkit (a set of techniques to assess asset vulnerabilities) to rank pipes according to risk magnitude and facilitating decisions about rehabilitation. Both studies aimed to improve the management of water supply systems, as well as minimizing failures and crises. It is important to note that both models depend on historical records of the network, such as failure rate and pipe age.

Finally, considering several studies related to the planning of pipe renewal in water distribution systems over a period of 10 years (2013/2022) and presented in Table 1, it was possible to find 50 criteria distributed in nine categories partially used in the presented models. Most of these studies associated hydraulic and mechanical criteria to pipe failures, but, they also presented a negligible representation to the water safety category. Many authors [3,45,48–52] highlight that pipe failures (breaks/leaks) may provoke water contamination, harming the health of the population. In addition, some limitations and difficulties for model implementation were identified, especially in small-scale distribution systems, such as: the use of statistical samples, mathematical estimates, complex regressions, historical data, programming techniques, and intensive computational resources. Therefore, it is essential to develop a simplified model that provides comprehensive criteria, including water quality, to guide the renewal of water distribution networks.

In this research, a risk assessment model of operational failures was designed in order to be applied in potable water distribution systems with restricted conditions (with imprecise data about the age of the infrastructure, and limited technical and financial resources). The model highlighted the pipes with the greatest risks within the system using as input hydraulic parameters and mechanical characteristics. These data allowed for the estimation of the probabilities of failures, and for the return of associated consequences, i.e., qualitative impact criteria reflected by the water quality index (WQI) and financial costs (associated to the replacement of the piping) as intermediate outputs. The risk of failure was consequently estimated using the probability of failure and the degree of the impact. Finally, the proposed model was applied to a real water distribution system located in the southeast part of the municipality of São José dos Campos, state of São Paulo, Brazil.

## 2. Materials and Methods

The methodology of this risk assessment model was structured for each pipe section in three main phases: (1) Determine the probability of failure in the pipe; (2) Evaluate the degree of impact due to the failure; and (3) Determine the risk of the failure.

### 2.1. Phase 1—Probability of Failure in the Pipe

In this stage, the goal was to determine the probability of failure of the pipelines in the water distribution network. To do this, we considered the hydraulic and mechanical properties of the pipes, such as pressure, velocity, flow rate, length, and diameter. These properties are essential for evaluating the risk of pipe failure and were referred to as pipe design parameters (PDPs).

For example, pressure is a crucial PDP in risk assessment, as the design of the water distribution network establishes the maximum and minimum pressure limits. The minimum limit aims to ensure adequate service to the points of consumption, while the maximum limit is related to the increase in the useful life of the tubes and especially, to the reduction of real losses in the pipelines. If the pressure exceeds the maximum limit, this can lead to the failure of the pipe. Similarly, excessive water velocity might corrode the inner walls of the pipes [53], while an inadequate flow rate can affect the service to the points of consumption [54]. Furthermore, longer pipes are more prone to failure due to pressure and mechanical stress, while pipes with smaller diameters are more prone to blockages and clogs.

To analyze the hydraulic and mechanical properties of the pipes, the QGIS software with the QGISRed v.015 [55] plugin and the Epanet [56] software were used. The plugin QGISRed allowed for the integration of the hydraulic modeling data from the Epanet with QGIS, simulating the behavior of water in the distribution system directly in the GIS. To determine these properties individually for each pipe, seven steps were followed: data acquisition (using six flow meters ultrasonic bulk water meter—Octave: 2 Ø150 M1/M3 and 4 Ø50 M2/M4–M6 integrated to Data-Logger's VT-490–Vectora: flow rate and pressure), network modeling, parameter definition, calibration, integration, results analysis, and documentation, described in detail in File S1.

For the calculation of the pipe pressure ($P_{Pipe}$), the average of the two corresponding nodes, the upstream ($P_i$) and downstream ($P_j$) node pressure, was considered, according to Equation (1)

$$P_{Pipe} = \frac{(P_i + P_j)}{2} \tag{1}$$

Once the hydraulic and mechanical parameters of the pipe have been determined, it receives an individual failure probability score ($FPS_i$) by associating its value to the respective range in Table 2, which illustrates the relationship between the PDPs and the pipe failure, which can occur in the form of quantitative events (e.g., pipe bursts) or quality failures (e.g., contamination of the pressurized flow) [37].

**Table 2.** Risk level and corresponding failure probability score.

| Pipe Design Parameters (PDPs) | | Risk Level and Corresponding Failure Probability Score ($FPS_i$) | | | | | | |
|---|---|---|---|---|---|---|---|---|
| | | V. Low 1 | Low 2 | R. Low 3 | Medium 4 | R. High 5 | High 6 | V. High 7 |
| Hydraulic [1] | $P_{min}$ (m.w.c) [2] | 25 | 25–22 | 22–19 | 19–16 | 16–13 | 13–10 | <10 |
| | $P_{max}$ (m.w.c) [2] | 25 | 25–30 | 30–35 | 35–40 | 40–45 | 45–50 | >50 |
| | $V_{min}$ (m/s) [3] | 1–0.9 | 0.9–0.8 | 0.8–0.7 | 0.7–0.6 | 0.6–0.5 | 0.5–0.4 | <0.4 |
| | $V_{max}$ (m/s) [3] | 0.9–1 | 1–1.1 | 1.1–1.2 | 1.2–1.3 | 1.3–1.4 | 1.4–1.5 | >1.5 |
| | Flow rate (L/s) | <0.8 | 0.8–21.6 | 21.6–42.5 | 42.5–63.3 | 63.3–84.2 | 84.2–105 | >105 |
| Mechanical | Length (m) | <10 | 10–108 | 108–206 | 206–304 | 304–402 | 402–500 | >500 |
| | Stiffness Ø (mm) | ≥300 | 300–250 | 250–200 | 200–150 | 150–100 | 100–50 | ≤50 |

Notes: Source: adapted by the authors based on Salehi et al. [30,37]. [1]—Values adjusted for the study area, according to normative limits [57], 50 mm ≤ nominal diameter (DN) ≤ 300 mm. [2]—$P_{min}$ > 25 m $H_2O$; $P_{max}$ < 25 m $H_2O$, it is considered a negligible risk (NR), i.e., $FPS_{Pressure}$ = 0. [3]—$V_{min}$ > 1 m/s; $V_{max}$ < 0.9 m/s, it is considered $FPS_{Velocity}$ = 0.

The minimum (min) and maximum (max) values of the PDPs are determined according to the valid references for the design and operation of urban water distribution networks [30,57,58]. However, to improve the classification of these parameters, adjustments are necessary depending on the characteristics of the study area [37]. In addition, the average values of these parameters were determined by taking five equal distances between the min-max values, resulting in seven classification categories (very low, low, relatively low, medium, relatively high, high, and very high) for each PDP.

The information obtained in this stage was essential to determine the failure probability score of the pipes and provided a solid foundation for the subsequent phases of this risk assessment model, as the FPS is a crucial factor to obtain the risk of failure.

### 2.2. Phase 2—Assessment of the Degree of Impact

In this stage, the objective was to determine the consequences related to potential failures in the water distribution network pipes, and evaluate the negative impacts that these failures may cause. The analysis focuses on two fundamental issues: water quality and the cost of pipe replacement.

Both water quality and the cost of pipe replacement are essential in the risk analysis, as a failure in the network can cause leaks and/or bursts, which might affect the quality of the water and/or increase water losses. This has a direct impact on the health of the population and on the budget of the company responsible for the system, factors that directly affect the socioeconomic development of the region.

The methodology used to determine the quality of the water and the cost of pipe replacement for the purpose of classifying the risk associated with these factors is presented in the following subitems.

2.2.1. Potable Water Quality

The water quality index (WQI) was necessary to determine the water quality in the distribution network, but it still required a geographic information system (GIS) to make its spatial representation in each pipe. Therefore, the process of obtaining and analyzing these results involved four consecutive steps: monitoring plan, analysis of potable water, definition and calculation of the WQI, and insertion of the data in the geographic information system (GIS), as detailed in File S2.

Once the WQI was determined individually in the pipe, keeping the same risk levels adopted in Table 2, an impact score was applied according to the values defined in Table 3. This process allowed for a more precise and comprehensive evaluation of the risk of pipe failure and its impacts on water quality.

**Table 3.** Risk level and corresponding impact score based on the water quality index.

| Impact Criteria | Risk Level and Corresponding Impact Score (IS$_{WQI}$) | | | | | | |
|---|---|---|---|---|---|---|---|
| | V. Low 1 | Low 2 | R. Low 3 | Medium 4 | R. High 5 | High 6 | V. High 7 |
| Water Quality Index CCME WQI$_{Pipe}$ [59] | 100–95 | 94–89 | 88–80 | 79–65 | 64–45 | 44–25 | 24–0 |

2.2.2. Pipe Replacement Cost

The water distribution networks are considered to be the most important and costly infrastructure assets in the system and their renewal is crucial to ensure a constant and reliable water supply [1,60,61]. As a result, it is fundamental to consider the financial consequences associated with pipe failure, since these actions involve high costs.

In this context, the estimated cost for pipe replacement was used as the financial criterion for assigning impact points. It is important to note that the costs associated with earth movement, ballast, slab, cradle, scaffolding, and pavement reconstruction were disregarded in the estimate, as it was assumed that the price of the material and the labor for the pipe laying represent all other costs related to the service.

The cost estimate for Brazil was based on the "Bank of Prices for Engineering Works and Services" [56], developed by the Basic Company Sanitation of the State of São Paulo, Brazil (SABESP). This price bank is widely disclosed and used as a reference, not only in the state of São Paulo, but it also has a high national influence in Brazil. Moreover, it is considered quite specific and meticulous for the area of basic sanitation. The values per linear meter for each pipeline in this study are presented in Table 4.

**Table 4.** Estimated cost of the distribution network pipes.

| Diameter (mm) | Abbreviation [1] | Price Per Linear Meter [2] | | | Pricing Source [3] | |
|---|---|---|---|---|---|---|
| | | Material | Labor | Total | Material | Labor |
| 50 | G.I | BRL 97.53 | BRL 24.18 | BRL 121.71 | HM01210 | 70,080,034 |
| 100 | | BRL 184.20 | BRL 28.19 | BRL 212.39 | HM01213 | 70,080,035 |
| 100 | CIP | BRL 342.25 | BRL 28.19 | BRL 370.44 | HM04118 | 70,080,035 |
| 200 | | BRL 441.19 | BRL 39.29 | BRL 480.48 | HM04105 | 70,080,035 |
| 300 | | BRL 656.56 | BRL 52.80 | BRL 709.36 | HM04107 | 70,080,039 |
| 50 | | BRL 97.53 | BRL 24.18 | BRL 121.71 | HM01210 | 70,080,034 |
| 60 | | BRL 150.09 | BRL 24.18 | BRL 174.27 | HM01211 | 70,080,034 |
| 85 | | BRL 331.56 | BRL 24.18 | BRL 355.74 | HM04129 | 70,080,034 |
| 125 | HDPE | BRL 94.47 | BRL 26.23 | BRL 120.70 | HM02085 | 70,140,034 |
| 90 | | BRL 64.19 | BRL 26.23 | BRL 90.42 | HM02093 | 70,140,034 |
| 100 | MPVC | BRL 103.41 | BRL 18.62 | BRL 122.03 | HM01930 | 70,080,003 |
| 125 | | BRL 103.41 | BRL 18.62 | BRL 122.03 | HM01930 | 70,080,003 |
| 150 | | BRL 182.55 | BRL 21.63 | BRL 204.18 | HM01931 | 70,080,004 |
| 200 | | BRL 323.24 | BRL 23.43 | BRL 346.67 | HM01932 | 70,080,005 |

**Table 4.** *Cont.*

| Diameter (mm) | Abbreviation [1] | Price Per Linear Meter [2] | | | Pricing Source [3] | |
|---|---|---|---|---|---|---|
| | | Material | Labor | Total | Material | Labor |
| 100 | | BRL 90.32 | BRL 18.62 | BRL 108.94 | HM01914 | 70,080,003 |
| 110 | | BRL 90.32 | BRL 18.62 | BRL 108.94 | HM01914 | 70,080,003 |
| 160 | | BRL 130.20 | BRL 21.63 | BRL 151.83 | HM07154 | 70,080,004 |
| 200 | | BRL 215.85 | BRL 23.43 | BRL 239.28 | HM07155 | 70,080,005 |
| 25 | | BRL 4.76 | BRL 2.41 | BRL 7.17 | HM01612 | 70,080,067 |
| 32 | PVC | BRL 9.53 | BRL 9.66 | BRL 19.19 | HM01613 | 70,080,023 |
| 40 | | BRL 16.55 | BRL 13.72 | BRL 30.27 | HM01614 | 70,080,012 |
| 50 | | BRL 19.02 | BRL 16.80 | BRL 35.82 | HM01615 | 70,080,001 |
| 60 | | BRL 25.84 | BRL 16.80 | BRL 42.64 | HM01915 | 70,080,001 |
| 75 | | BRL 51.64 | BRL 17.39 | BRL 69.03 | HM01617 | 70,080,002 |
| 85 | | BRL 55.30 | BRL 17.39 | BRL 72.69 | HM01916 | 70,080,002 |
| 50 | PVC SSR | BRL 25.84 | BRL 16.80 | BRL 42.64 | HM01915 | 70,080,001 |
| 75 | | BRL 55.30 | BRL 17.39 | BRL 72.69 | HM01916 | 70,080,002 |

Notes: [1] materials: galvanized iron pipe (G.I); cast iron pipe (CIP); high-density polyethylene (HDPE); modified polyvinyl chloride (MPVC); polyvinyl chloride (PVC); and polyvinyl chloride with spigot/socket/ring (PVC SSR). [2] 1 USD = 54,060 BRL on 30 September 2022, according to the Brazilian Central Bank. [3] Basic Sanitation Company of the State of São Paulo (Sabesp) (September, 2022) [62].

To account for the costs associated with the rehabilitation of each pipe in the water distribution network, the length is multiplied by the associated price from Table 4.

Once the value of the pipe is determined, it receives an impact score ($IS_{COST}$), as indicated in Table 5. By incorporating network repair expenses, in this risk assessment model as a final consequence of failure, the result, in addition to being an indicator to prioritize renewal actions, allows managers to estimate the total cost of a pipe replacement project by using pricing strategies that consider the resulting value. In this way, it is possible to price other costs related to the completion of the services.

**Table 5.** Scoring criteria for pipe repair ($IS_{COST}$).

| Impact Criteria | | Risk Level and Corresponding Impact Score ($IS_{COST}$) | | | | | | |
|---|---|---|---|---|---|---|---|---|
| | | V. Low 1 | Low 2 | R. Low 3 | Medium 4 | R. High 5 | High 6 | V. High 7 |
| Financial | Cost Range (1.000 × BRL) | 0–2 | 2–19 | 19–36 | 36–53 | 53–70 | 70–87 | >87 |

### 2.3. Phase 3—Risk of Failure

The final stage of the model is to determine the overall risk of failure for each pipe. The risk of failure combines the cumulative score of the probability of failure (total failure probability score—$TFP_S$) presented in Equation (2) and the qualitative and financial consequences (total impact score—$IS_{Total}$) presented in Equation (3). The risk of failure (RFS) is finally defined by Equation (4).

$$TFP_S = FPS_{Pmin} + FPS_{Pmax} + FPS_{Vmin} + FPS_{Vmax} + FPS_{FR} + FPS_\theta + FPS_L \tag{2}$$

$$IS_{Total} = IS_{WQI} + IS_{COST} \tag{3}$$

$$RFS = TFP_S \times IS_{Total} \tag{4}$$

The scores for each probability of failure and the consequences are added for each pipeline segment, the product of these sums results in the value of the risk of failure. The result serves as a comparison mechanism between failures, in this case, a dimensionless number for each pipe that can then be compared to all others, serving as a classification system to highlight the highest risk within a water distribution network. The risks of failure scores (RFS) have a range from 12 to 686 and were subdivided into four groups: low (15%: 12–69), medium (30%: 70–184), high (30%: 185–396), and very high (25%: 400–686), each with its own color coding and suggested actions, as shown in Tables 6 and 7.

**Table 6.** Risk matrix—RFS values.

| IS$_{Total}$ / IS$_{Total}$ | Total Failure Probability Score | | | | | | | | | | | | | | | | | | | | | | | | | | | | | | | | | | | | | | | | | | | |
|---|---|---|---|---|---|---|---|---|---|---|---|---|---|---|---|---|---|---|---|---|---|---|---|---|---|---|---|---|---|---|---|---|---|---|---|---|---|---|---|---|---|---|---|
| | 6 | 7 | 8 | 9 | 10 | 11 | 12 | 13 | 14 | 15 | 16 | 17 | 18 | 19 | 20 | 21 | 22 | 23 | 24 | 25 | 26 | 27 | 28 | 29 | 30 | 31 | 32 | 33 | 34 | 35 | 36 | 37 | 38 | 39 | 40 | 41 | 42 | 43 | 44 | 45 | 46 | 47 | 48 | 49 |
| 2 | 12 | 14 | 16 | 18 | 20 | 22 | 24 | 26 | 28 | 30 | 32 | 34 | 36 | 38 | 40 | 42 | 44 | 46 | 48 | 50 | 52 | 54 | 56 | 58 | 60 | 62 | 64 | 66 | 68 | 70 | 72 | 74 | 76 | 78 | 80 | 82 | 84 | 86 | 88 | 90 | 92 | 94 | 96 | 98 |
| 3 | 18 | 21 | 24 | 27 | 30 | 33 | 36 | 39 | 42 | 45 | 48 | 51 | 54 | 57 | 60 | 63 | 66 | 69 | 72 | 75 | 78 | 81 | 84 | 87 | 90 | 93 | 96 | 99 | 102 | 105 | 108 | 111 | 114 | 117 | 120 | 123 | 126 | 129 | 132 | 135 | 138 | 141 | 144 | 147 |
| 4 | 24 | 28 | 32 | 36 | 40 | 44 | 48 | 52 | 56 | 60 | 64 | 68 | 72 | 76 | 80 | 84 | 88 | 92 | 96 | 100 | 104 | 108 | 112 | 116 | 120 | 124 | 128 | 132 | 136 | 140 | 144 | 148 | 152 | 156 | 160 | 164 | 168 | 172 | 176 | 180 | 184 | 188 | 192 | 196 |
| 5 | 30 | 35 | 40 | 45 | 50 | 55 | 60 | 65 | 70 | 75 | 80 | 85 | 90 | 95 | 100 | 105 | 110 | 115 | 120 | 125 | 130 | 135 | 140 | 145 | 150 | 155 | 160 | 165 | 170 | 175 | 180 | 185 | 190 | 195 | 200 | 205 | 210 | 215 | 220 | 225 | 230 | 235 | 240 | 245 |
| 6 | 36 | 42 | 48 | 54 | 60 | 66 | 72 | 78 | 84 | 90 | 96 | 102 | 108 | 114 | 120 | 126 | 132 | 138 | 144 | 150 | 156 | 162 | 168 | 174 | 180 | 186 | 192 | 198 | 204 | 210 | 216 | 222 | 228 | 234 | 240 | 246 | 252 | 258 | 264 | 270 | 276 | 282 | 288 | 294 |
| 7 | 42 | 49 | 56 | 63 | 70 | 77 | 84 | 91 | 98 | 105 | 112 | 119 | 126 | 133 | 140 | 147 | 154 | 161 | 168 | 175 | 182 | 189 | 196 | 203 | 210 | 217 | 224 | 231 | 238 | 245 | 252 | 259 | 266 | 273 | 280 | 287 | 294 | 301 | 308 | 315 | 322 | 329 | 336 | 343 |
| 8 | 48 | 56 | 64 | 72 | 80 | 88 | 96 | 104 | 112 | 120 | 128 | 136 | 144 | 152 | 160 | 168 | 176 | 184 | 192 | 200 | 208 | 216 | 224 | 232 | 240 | 248 | 256 | 264 | 272 | 280 | 288 | 296 | 304 | 312 | 320 | 328 | 336 | 344 | 352 | 360 | 368 | 376 | 384 | 392 |
| 9 | 54 | 63 | 72 | 81 | 90 | 99 | 108 | 117 | 126 | 135 | 144 | 153 | 162 | 171 | 180 | 189 | 198 | 207 | 216 | 225 | 234 | 243 | 252 | 261 | 270 | 279 | 288 | 297 | 306 | 315 | 324 | 333 | 342 | 351 | 360 | 369 | 378 | 387 | 396 | 405 | 414 | 423 | 432 | 441 |
| 10 | 60 | 70 | 80 | 90 | 100 | 110 | 120 | 130 | 140 | 150 | 160 | 170 | 180 | 190 | 200 | 210 | 220 | 230 | 240 | 250 | 260 | 270 | 280 | 290 | 300 | 310 | 320 | 330 | 340 | 350 | 360 | 370 | 380 | 390 | 400 | 410 | 420 | 430 | 440 | 450 | 460 | 470 | 480 | 490 |
| 11 | 66 | 77 | 88 | 99 | 110 | 121 | 132 | 143 | 154 | 165 | 176 | 187 | 198 | 209 | 220 | 231 | 242 | 253 | 264 | 275 | 286 | 297 | 308 | 319 | 330 | 341 | 352 | 363 | 374 | 385 | 396 | 407 | 418 | 429 | 440 | 451 | 462 | 473 | 484 | 495 | 506 | 517 | 528 | 539 |
| 12 | 72 | 84 | 96 | 108 | 120 | 132 | 144 | 156 | 168 | 180 | 192 | 204 | 216 | 228 | 240 | 252 | 264 | 276 | 288 | 300 | 312 | 324 | 336 | 348 | 360 | 372 | 384 | 396 | 408 | 420 | 432 | 444 | 456 | 468 | 480 | 492 | 504 | 516 | 528 | 540 | 552 | 564 | 576 | 588 |
| 13 | 78 | 91 | 104 | 117 | 130 | 143 | 156 | 169 | 182 | 195 | 208 | 221 | 234 | 247 | 260 | 273 | 286 | 299 | 312 | 325 | 338 | 351 | 364 | 377 | 390 | 403 | 416 | 429 | 442 | 455 | 468 | 481 | 494 | 507 | 520 | 533 | 546 | 559 | 572 | 585 | 598 | 611 | 624 | 637 |
| 14 | 84 | 98 | 112 | 126 | 140 | 154 | 168 | 182 | 196 | 210 | 224 | 238 | 252 | 266 | 280 | 294 | 308 | 322 | 336 | 350 | 364 | 378 | 392 | 406 | 420 | 434 | 448 | 462 | 476 | 490 | 504 | 518 | 532 | 546 | 560 | 574 | 588 | 602 | 616 | 630 | 644 | 658 | 672 | 686 |

*Total Impact Score* (row axis)

**Table 7.** Legend of the failure risk scores and linguistic scale.

| RFS Value | Color [1] | | Failure Risk Level | Suggested Action [2] |
| :---: | :---: | :---: | :---: | :---: |
| | RGB | Description | | |
| ≤69 | (0, 158, 15) | Green | Low | A |
| 70 to 184 | (86, 180, 233) | Blue | Medium | B |
| 185 to 396 | (240, 228, 66) | Yellow | High | C |
| ≥400 | (230, 159, 0) | Orange | Very High | D |

Notes: [1] the adaptation of the color palette was chosen to better identify it with the basic map colors and to avoid any misunderstandings in color identification by potential visually impaired users or those suffering from color blindness. [63]. [2] Action: A = routine inspection and condition assessment after 10 years; B = condition assessment/repair will be necessary in the next 10 years; C = condition assessment/rehabilitation will be necessary in the next 2 years; D = condition assessment/replacement may be necessary within 6 months.

Finally, with this numerical score, the classification and creation of the "failure risk" map of the water distribution network pipes can be applied.

## 3. Case Study

The developed model was applied to the water supply system (WSS) of the Aerospace Science and Technology Department (DCTA), located in São José dos Campos, São Paulo, Brazil.

In this water supply system (Figure 1), raw water is collected from underground sources (three tubular wells) to the disinfection unit of the water treatment station. Additionally, a surface source (Vidoca Stream) offers raw water conducted through supply lines to the water treatment station (WTS), which is a conventional station that operates by gravity and includes coagulation, flocculation, sedimentation, rapid filtration, and chlorination processes. The potable water storage capacity at the WTS is 2500 m$^3$. Then, treated water is supplied to the consumption points through the distribution network, which covers a total area of 7.2 km$^2$. The network is composed of pipelines of various diameters (from 25 mm to 300 mm) and materials presented in Table 4 (G.I, PVC, MPVC, CIP, and HDPE). The WTS produces an average of 3028 m$^3$/day of potable water and serves approximately 20,000 people, 5000 of which are permanent and 15,000 are transient. It is important to highlight that a global improvement in the water quality index after some intervention actions carried out from 2016 to 2018 in this water supply system (WSS) motivated this study.

Currently, the operating distribution network totals 38,677 m and has seventeen reservoirs. Ten reservoirs are located in the low zone (altitude < 610 m): R1 to R10, while four are located in the high zone (altitude > 610 m): R11 to R14. In the high zone, there is also a peak-demand reservoir, R15, responsible for maintaining supply during peak hours. One reservoir (R16) is responsible for supplying the high zone reserve subsystem. Finally, the reservoir located at the water treatment station, R17, supply all of the reserve subsystems.

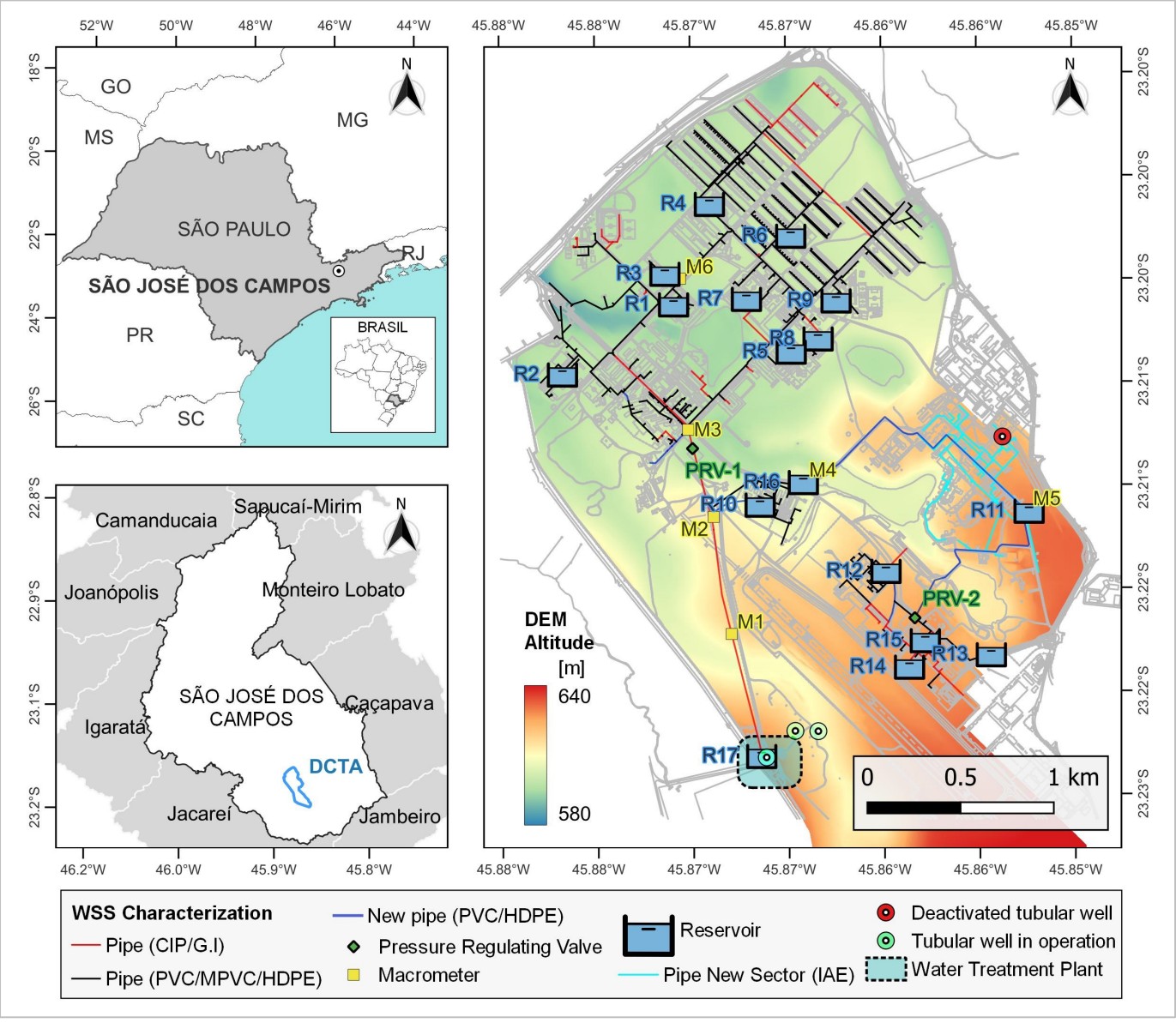

**Figure 1.** Location and characterization of the water distribution system.

## 4. Results

It is important to note that the values depended, respectively, on the applied WQI and the feasible costs of the region (country, state, and city). Therefore, the numerical parameters of these tables (2, 3, and 5) should not be treated as absolute values for application in another water distribution system. Finally, the individual representation of each of the criteria (hydraulic, mechanical, qualitative, and financial) was presented in the GIS to support managers to decide what preventive (reduce the probability of the occurrence of an event) and/or protective (mitigate the consequences) actions should be prioritized.

### 4.1. Phase 1—Probability of Failure

The points of failure probability scores ($FPS_i$) of the hydraulic and mechanical parameters were calculated using Table 2 and the results of the hydraulic simulation in the GIS with the QGISRed plugin, which integrated the hydraulic model produced in the Epanet. These results were used to create maps of the "risk level" related to hydraulic and mechanical failures in pipes, as shown in Figures 2 and 3.

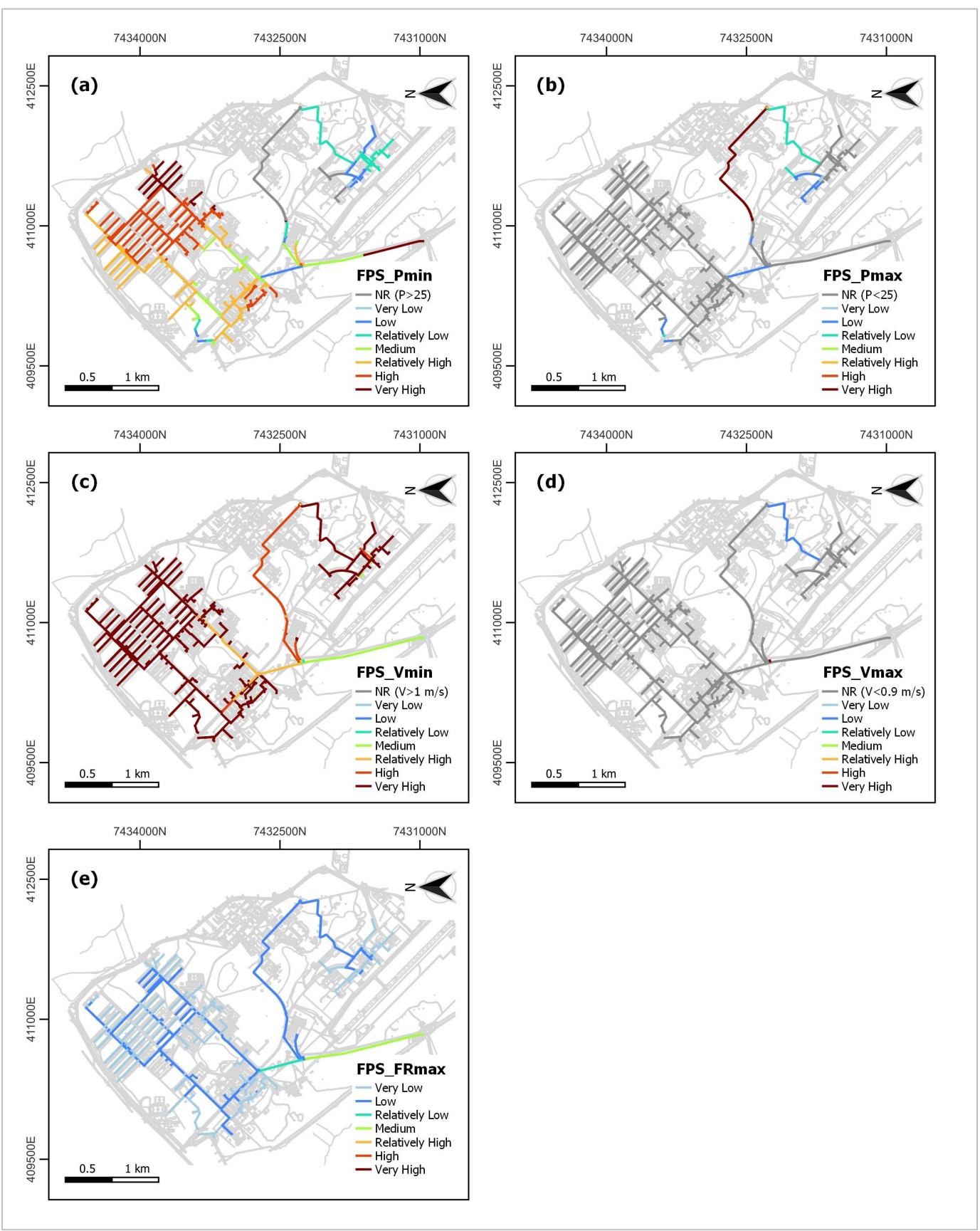

**Figure 2.** Risk level associated with hydraulic failure in pipes: (**a**) minimum pressure; (**b**) maximum pressure; (**c**) minimum velocity; (**d**) maximum velocity; and (**e**) maximum flow.

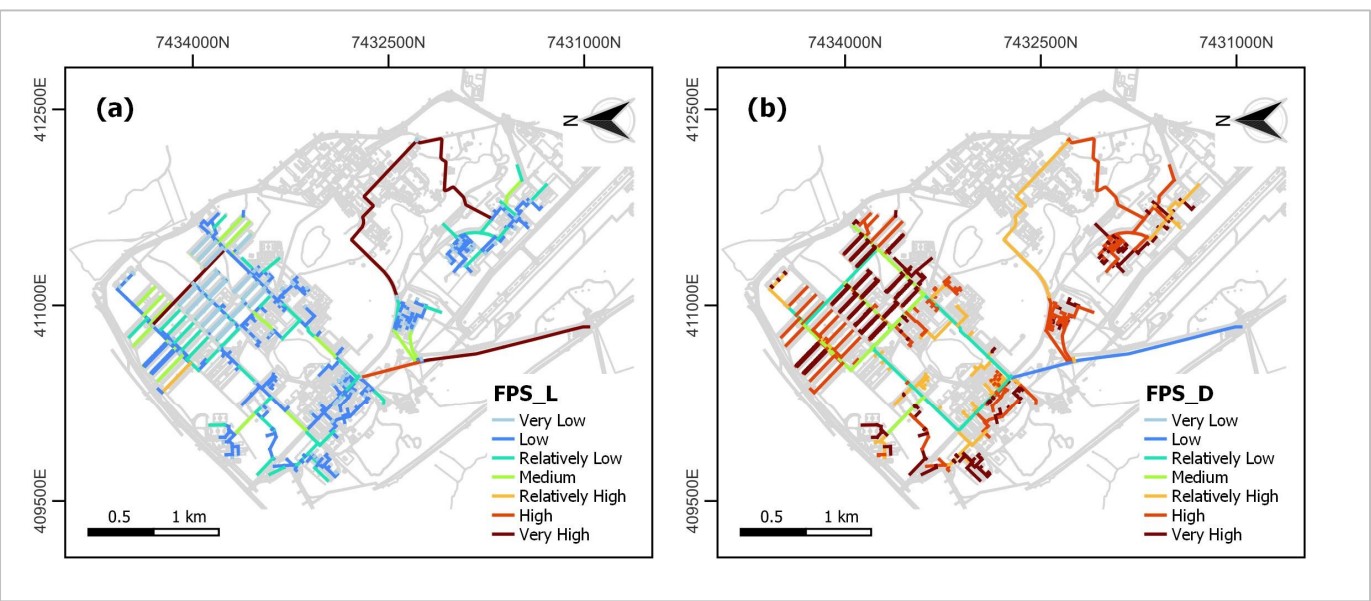

**Figure 3.** Risk level associated with mechanical failure in pipes: (**a**) length and (**b**) diameter.

For the design of the water distribution network, the minimum dynamic pressure ($P_{min}$) limits and maximum static pressure ($P_{max}$) limits are established. The $P_{min}$ limit aims to ensure adequate service at the points of consumption, while the $P_{max}$ limit is related to the resistance of the pipelines and physical losses. According to [57], $P_{min}$ must be 10 m $H_2O$ and $P_{max}$ must be 50 m $H_2O$. For $P_{min}$ (Figure 2a), the "very high risk" sections (in the beginning and end of network) are below the recommended limits, while the remaining pipe segments meet the required standards, with risk levels between "high" and "very low". Analyzing the results of Figure 2b, it can be stated that there is only a network stretch related to $P_{max}$ (Figure 2b), with most risk levels between "low" and negligible risk.

The minimum velocity limit ($V_{min}$) is recommended to ensure a permanent water flow in the distribution network so as not to affect the quality of the water supplied to the consumer [64]. Low velocities (above the minimum limit) in the distribution network favor durability, in terms of pipe abrasion, while higher velocities reduce the diameters and thus the cost of purchasing and installing the pipes. Moreover, velocities above the maximum limit may cause noise in the pipes and favor wear and tear due to abrasion and cavitation, increasing maintenance costs [65]. NBR 12218/2017 establishes that the maximum velocities ($V_{max}$) must be limited to a head loss of 10 m/km and that velocities lower than 0.40 m/s should be avoided [57]. Thus, it can be stated that low velocities might be affecting the quality of water in the system, especially in end-of-network pipes with a "very high" risk (Figure 2c), while values above the recommended standards were not observed for $V_{max}$ (Figure 2d).

The maximum flow rate limit ($FR_{max}$) is related to $V_{max}$ and the roughness of the pipe, and it must meet the maximum daily demands of the water distribution network's consumption points [58]. The analysis of the $FR_{max}$ result (Figure 2e) shows that the operating distribution network is oversized for the current consumption standard, with a risk level ranging from "medium" to "very low".

The structural deterioration of the pipe, associated with failure, was evaluated through two factors: diameter and length. The first is considered vital for failures in the distribution network and is reported in millimeters (mm), and the risk is inversely proportional to the diameter, as it is expected that the pipe with a larger diameter has a thicker wall compared to another pipe with a smaller diameter [14,66]. The second has been considered as one of the basic static parameters for structural integrity and is measured in meters (m), that is, the risk is directly proportional to the length, as it is expected that longer pipe segments are more susceptible to ruptures or leaks caused by external events (e.g., soil, loading,

urbanization, etc.) [33,67]. Thus, it can be stated from the analysis of Figure 3 that there are many more pipelines susceptible to mechanical failure derived from the diameter (Figure 3b) than from the length (Figure 3a) due to a high and very high-risk classifications. However, the impacts of a failure due to the length are more damaging, as the pipelines with high risk are the main water distribution system feeders.

### 4.2. Phase 2—Assessment of the Degree of Impact

The qualitative and financial scores of the degree of impact of the individual criteria (IS) described in item 2.2 are presented in the following subitems. Tables 3 and 5 were used in conjunction with geoprocessing and spatial analysis tools to determine the impact scores. These resources were employed in the calculation of the quality of potable water ($IS_{WQI}$) and the estimated repair cost ($IS_{COST}$), and results were obtained for each pipe in the water distribution network.

#### 4.2.1. Potable Water Quality

The spatial representation using the inverse distance weighting (IDW) of the results of the water quality index modified from the Council of Canadian Ministries of the Environment ($CCMEWQI_{Mod}$) along with the use of geospatial tools, detailed in File S2, made it possible to obtain the water quality in each distribution network pipe, and thus determine the $IS_{WQI}$ individually, as illustrated in Figure 4.

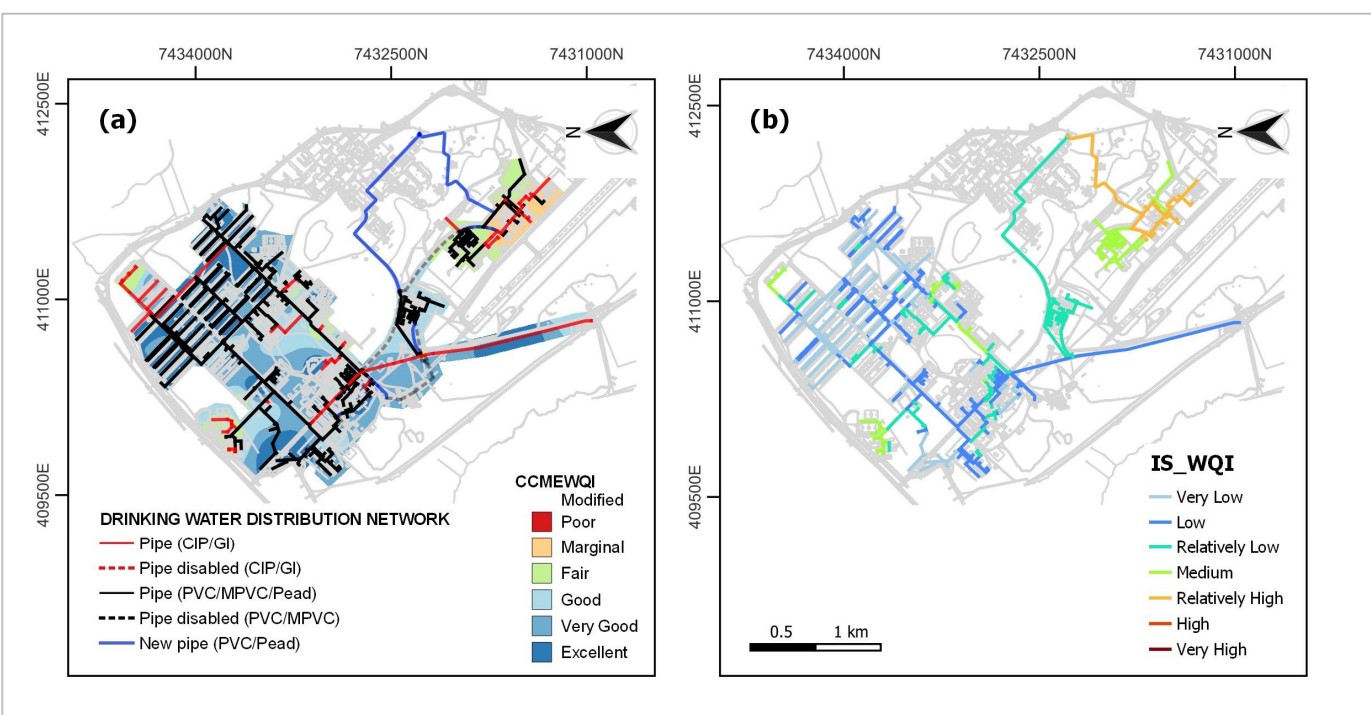

**Figure 4.** Qualitative/social consequence: (**a**) IDW interpolation—water quality; (**b**) ISWQI impact degree.

Analyzing the spatial distribution of the $CCMEWQI_{Mod}$, it can be seen that the worst index values are concentrated in the locations served by metal pipes (CIP and G.I), with an increase in the high zone (altitude > 610 m) and at the end of the network; therefore, the impact degree of these pipes presents a higher risk. In general, the impact degree due to variations in water quality in the system ranged from "very low" to "relatively high".

#### 4.2.2. Pipe Replacement Cost

A simple field calculation was implemented to calculate the estimated repair cost of each pipe in the distribution system. This involved multiplying the linear meter price

(Table 4) of each pipe segment by its total length. Once the value in the pipe was determined, it received an impact score ($IS_{COST}$) by applying its index value in Table 5 for the risk classification, as illustrated in Figure 5. The analysis of these results allowed to identify the distribution network pipes with the highest financial impact associated with their replacement. In general, the highest scores are concentrated in the main pipes that supply the low and high zones, followed by large diameters and/or length pipes, with a consequence of failure between "very high to relatively high". The remaining pipes, the more representative part of the network, had a consequence of failure between "medium and very low".

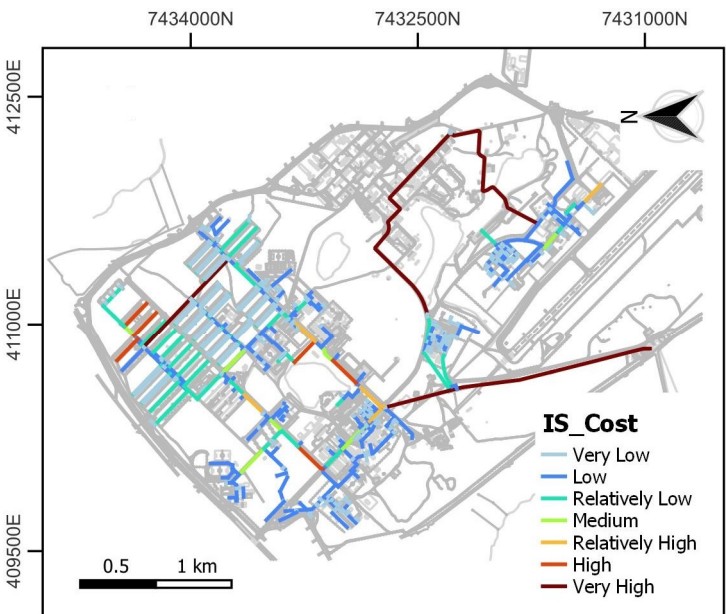

**Figure 5.** Financial consequence: pipe replacement cost.

Although the methodology provided an estimate of the financial costs associated with the replacement of each pipe in the water distribution network, it is essential that decisions regarding the necessary materials and services be defined through more detailed engineering work, taking into account undesirable design parameters presented by GIS and other factors, such as demand growth, environment, and modern construction techniques that may impact the total cost of the project. The choice of actions to be taken must be based on the specific needs of the water supply system and the best engineering practices available to ensure the efficiency and safety of the network, considering technical, economic, and environmental factors.

### 4.3. Phase 3—Risk of Failure

The final stage in the application of the model involved calculating the probability of failure score of the design parameters "PDPs" and total impact "$IS_{TOTAL}$" for each pipe segment, and then using Tables 6 and 7 to determine the risk of failure score (RFS). A simple field calculation was performed in QGIS to automatically insert the RFS (Equation (4)) values into the attribute table of the distribution network for each pipe.

The final result is an attribute column dedicated exclusively to the RFS value for each pipe segment in the water distribution system. The RFS values of all 1217 individual pipe segments in the network were then categorized and represented in the GIS to visually illustrate the priority and risk level of each pipe, as shown in Figure 6.

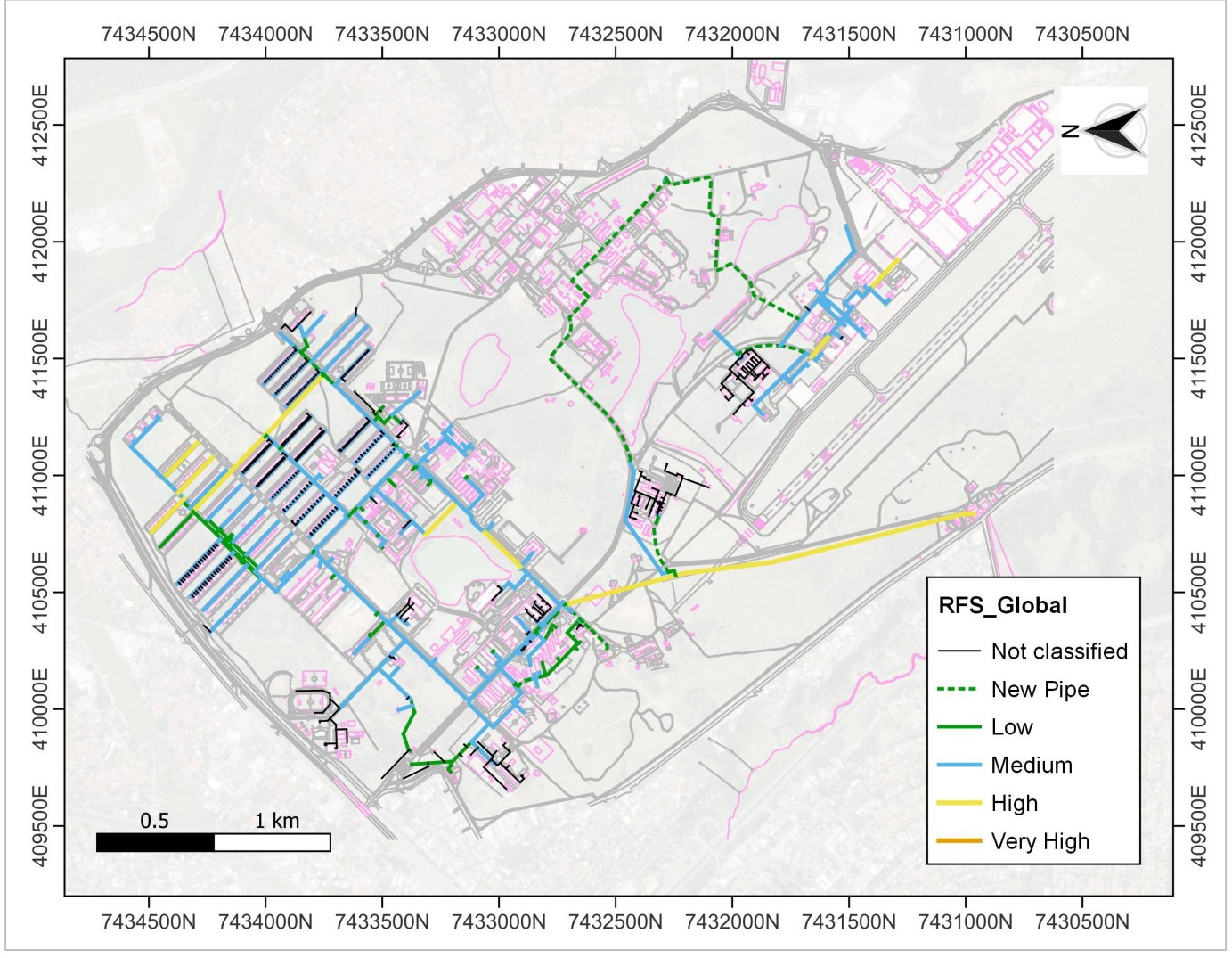

**Figure 6.** Water distribution network failure risk map.

It is important to note that due to the lack of some operational data, micro-metering readings, and the monitoring of the level of reservoirs in the low zone, the hydraulic modeling of this distribution system was simplified, as detailed in File S1. Therefore, some secondary network pipelines were represented in the RFS map as "unclassified" (black color). For the new distribution networks installed and those that were replaced between 2019 and 2021, the RFS classification of the pipes was manually altered to "new pipes" (dashed green color) to avoid prioritizing unnecessary improvement actions regarding the rehabilitation of the distribution network pipes.

In this water distribution network, no pipes were identified with RFS values equal to or greater than 313 (orange color), that is, a "very high" failure risk.

The pipes with the highest risk of failure in this system have RFS values between 162 and 312 (yellow color), that is, a "high" failure risk; therefore, they are a priority for evaluation and renewal in the next 2 years. In this range, there are 11 network segments, 10 cast iron pipes (CIP) that total 3690 m in length and with diameters of 85, 100, and 300 mm, and only one modified polyvinyl chloride pipe (MPVC) with 210 m in length and 200 mm in diameter. The total estimated cost for replacing these networks was BRL 3,227,479.70.

The remaining pipes had RFS values between 70 and 161 (blue color) or scores equal to or less than 69 (green color). These pipes were considered with the lowest chance of failure compared to all other segments of the hydraulic network.

## 5. Discussion

The results classified 77% of the total length of the distribution system ($\approx$30 km) to be considered for model validation, highlighting 11 pipes with a high risk of failure ($\approx$3.7 km) and an estimated replacement value of BRL 3.2 million. Although 23% of the total network length was not classified by the RFS index due to the simplification of the hydraulic model that limited the obtaining of necessary hydraulic parameters, these pipes represent a lower risk to the system, since they are secondary networks of reduced length. In addition, partial results obtained regarding mechanical, qualitative, and financial parameters can be considered individually or collectively to prioritize interventions in these pipes, adapting the risk matrix to available results. For example, when evaluating the impact on potable water quality individually, the worst results (higher risk) were observed in locations served by metal pipes (CIP and G.I) aggravated in the high zone and in end-of-network sections; therefore, priority for replacement or rehabilitation of these secondary pipes should be prioritized.

The availability of data necessary for obtaining the project parameters and water quality in each pipe is a critical factor for this model. However, this information can be obtained at any time, unlike models that require historical records of infrastructure and/or operational failures in the network, which if not identified or are inaccurate, can make their application unfeasible or result in inefficient planning [1,34,35,37]. Additionally, the cost of pipe replacement should be improved by incorporating other costs associated with the performance of the services to avoid any distortions in the results.

It is recommended that this risk assessment model be applied to different water distribution systems for comparison with other renewal methods and for improvement. Although the results obtained in a real case study have presented the potential of the model to assist the planning of improvement actions in a system, the solutions may be different if applied in other locations. Additionally, it is essential to regularly update the design parameters and water quality using field data and financial values for tube replacement to prioritize renewal activities in future years based on the updated RFS index.

Finally, to turn the current model into a predictive model instead of its current casual structure for prioritization, it is advisable to incorporate systematic data recording (hydraulic, mechanical, qualitative, and financial) using, for example, the supervisory control and data acquisition (SCADA) system integrating macro-measurements into micro-measurements. The proposed method is also flexible to include other contributing factors and consequences, as well as incorporate priorities defined by the involved water concessionaires.

## 6. Conclusions

The proposed model was able to provide accurate information about the pipes (material and diameter for primary and secondary tubes), visually identifying the most vulnerable, sensitive, and high-risk parts within the system. This model was flexible in dealing with missing data and the spatial representation of the water quality index (WQI) was an important tool for monitoring and controlling the potable water distributed to the population, preventing health risks.

For large water distribution systems, it is recommended to include other relevant criteria (contributing factors and consequences) and systematically record the required data to improve the accuracy of the results. In this way, a more complete and accurate analysis of the network situation and the necessary actions to ensure the quality of the distributed water may be obtained.

Moreover, for small- and medium-sized water distribution systems with technical and financial limitations, this risk assessment model can be highly advantageous as it uses freely accessible computer tools and a simple methodology, without the need for complex statistical models, mathematical estimates, advanced regressions, or intensive computational resources. Additionally, it provides the essential results for the development of maintenance and rehabilitation schedules for the network.

**Supplementary Materials:** The following supporting information can be downloaded at: https://www.mdpi.com/article/10.3390/w15081509/s1, File S1: Hydraulic Modeling [55,56,68–72]; File S2: Potable Water Quality [59,73–91].

**Author Contributions:** Conceptualization, R.N. and M.P.; methodology, R.N.; validation, R.N.; formal analysis, M.P.; investigation, R.N.; resources, M.P.; data curation, R.N. and E.A.; writing—original draft preparation, R.N.; writing—review and editing, M.P.; visualization, R.N. and E.A.; supervision, M.P. All authors have read and agreed to the published version of the manuscript.

**Funding:** The authors are grateful for the financial support received by Coordenação de Aperfeiçoamento de Pessoal de Nível Superior (CAPES) and Simão & Simão-Construtora e Incorporadora.

**Data Availability Statement:** The datasets used and/or analyzed during the current study are available from the corresponding author upon reasonable request.

**Acknowledgments:** To the Aerospace Science and Technology Department (DCTA) and the São Paulo State Basic Sanitation Company (SABESP) for their valuable support in the availability of the technical and operational data of the water distribution system and the Engineering Works and Services Price Bank, respectively. Their support was essential for the completion of this research.

**Conflicts of Interest:** The authors declare no conflict of interest.

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
