# Peer review of "Risk Assessment Model for the Renewal of Water Distribution Networks: A Practical Approach"

_water, doi:10.3390/w15081509_

Round 1

Reviewer 1 Report

Good review of work carried out. The authors have carried out detailed study on the water network and I commend on their work. The following are my suggestions and questions.

Reduce the number of references. Perhaps limit to only the last 10 years or so if possible. Need to comment on the references, instead of bundling multiple references for the same point.

It is better to not use sentences in the first person, such as using "we".

Table A2. There is a large difference between the two consumption values. To what do the authors contribute this discrepancy to?

Author Response

I would like to thank you for the comments and suggestions received during the review of our manuscript titled "Risk Assessment Model for Renewal of Water Distribution Networks. Here are our changes:

  1. We considered the last 10 years for references related to risk assessment models (lines: 76, 561 - 613). We commented most important articles related to the work.
  2. We corrected phrases written in the "first person".
  3. Below is our justification for the difference between the two consumption values presented in Table A2. We also explained it better in the appendix A (pg. 25 of file between tables A1 and A2).

 “With the simplification of the hydraulic model, secondary networks were disregarded, and the number of reservoirs was reduced. Consequently, the consumption of these customers was considered at the NODE representing the water inlet to the variable level reservoir (RNV), which was removed from the model. The node supplies the region and/or the sum of the volume consumed by each customer through the same distribution network. It should be emphasized that the base consumption value assigned at the NODE was provided by the system operator and adjusted based on the flow rate recorded by the flow meter in the respective region (MM1, MM2, MM3, MM4, MM5, and MM6) due to the absence of micro-metering (water meters) for all customers. In other words, since it was not possible to perform a water balance between micro-metering and macro-metering, losses were incorporated into consumption for system calibration, as detailed in Table A2 and represented in Figure A3.”

We hope that the changes made and the responses provided adequately address the reviewers' comments. We look forward to hearing your response and are available to provide additional information or make further changes if necessary.

Reviewer 2 Report

The authors presented a risk assessment model for water distribution pipeline systems. The proposed model uses a simple methodology that does not depend on statistical models, mathematical estimates, complex regressions, or intensive computational resources. A real case study was applied to show the usage and performance. The manuscript was well organized and basically well presented, including comprehensive references and introductions to the methods.

1.      More recent references are encouraged to be cited in the introduction about “Physical models”. For example, “A transient-features-based diagnostic method of multi incipient cracks in pipeline systems” (2022), “Influencing mechanisms of gas bubbles on propagation characteristics of leakage acoustic waves in the gas-liquid two-phase flow” (2023), “Transient simulation and diagnosis of partial blockage in long-distance water supply pipeline systems” (2021), etc.

2.      Please add references and simply explain why for the following statement:

“excessive water velocity can damage the inner walls of the pipes, while inadequate flow rate can affect service to the points of consumption.”

3.      Although the methods seem to be reasonable, is it possible to validate the model through a historic analysis? Otherwise, the results cannot be that convincing after all.

4.      There are some minor typing/grammar errors, please carefully check through.

Overall, I would like to suggest a major revision before consideration of publication.

Author Response

I would like to thank you for the comments and suggestions received during the review of our manuscript titled "Risk Assessment Model for Renewal of Water Distribution Networks: A Practical Approach." We have carefully worked on the revisions and believe that the manuscript has significantly improved. Below, we provide details about the main changes made and how we addressed the reviewers' comments.

  1. We cited the most recent references suggested by the reviewer in the introduction about "Physical Models (lines 46-57)" and also replaced older articles by newer ones (lines 561-616).
  2. The references related to the text "excessive water velocity can damage the inner walls of the pipes, while inadequate flow ratecan affect service to the points of consumption." were inserted in the revised manuscript (lines 185-188) and our whole explanation follows below:

 “Excessive water velocity can cause various issues within the pipes. High velocities can lead to erosion and corrosion of the inner walls, which can weaken the pipe structure and eventually result in pipe failures, such as leaks or bursts. Additionally, high water velocities can cause water hammer, a sudden pressure surge that occurs when the flow of water is abruptly stopped or changed. Water hammer can produce substantial pressure fluctuations, further damaging the pipe system and its components. On the other hand, inadequate flow rate can also have negative consequences. Low flow rates can result in insufficient water pressure at the points of consumption, affecting the quality of service for end-users. Insufficient flow can also lead to water stagnation within the pipes, increasing the risk of microbial growth and the potential for waterborne diseases. Furthermore, stagnant water may cause a buildup of sediments and biofilms, which can deteriorate water quality and clog the pipes, ultimately requiring costly maintenance and repair interventions. Therefore, it is crucial to maintain a balance between water velocity and flow rate within the distribution network to ensure the durability of the pipe system and provide reliable service to consumers. The main relevant factors related to the quality of water were inserted in the main text (lines 185-188).

     3. Regarding the reasonableness of the method and validation of the model through a historical analysis, we clarify:

“Risk assessment models for the planning of renewal and/or rehabilitation in water distribution networks often rely on historical data. These data allow for the identification of patterns and trends related to failures, facilitating the understanding of causes and effects within the system, and enabling the prioritization of improvement actions and optimization of resource allocation.

Historical data are essential for calibrating and validating risk assessment models, increasing confidence in predictions and ensuring the effectiveness of renewal and rehabilitation actions. The method proposed in this study, like others in the literature, has limitations (discussed in item 5). However, most of the required information can be obtained within a few years through monitoring hydraulic parameters (necessary for hydraulic modeling) and water quality parameters (physical-chemical and microbiological parameters for the calculation of the Water Quality Index - WQI).

In this context, the historical analyses applied in the conceived risk assessment model focus on water quality data, highlighting the methodology's potential by considering the WQI as a direct consequence of pipe failures. Figure N1 (inserted on page 51 of the attached file) represents the improvement actions (pipe replacement and leak repairs) implemented in the system (2019/2021), while Figure N2 (inserted on page 52 of the attached file)  shows the effects on water quality, considering the previous three-year period (2016/2018). Although the improvement actions took place before the conception of this risk assessment model, these results (Figure N2) motivated its development and future research for its refinement.

Additionally, Figures N1 and N2 show that WQI naturally indicates the influence of the state of pipes in the distributed water. We included lines 331-333 to ilustrate the fundamentals of the work.

Round 2

Reviewer 2 Report

The authors have addressed all my concerns.